# Long-Term Survival Prediction Model for Elderly Community Members Using a Deep Learning Method

**DOI:** 10.3390/geriatrics8050105

**Published:** 2023-10-23

**Authors:** Kyoung Hee Cho, Jong-Min Paek, Kwang-Man Ko

**Affiliations:** 1Department of Health Policy and Management, SangJi University, Wonju-si 26339, Republic of Korea; chokh017@sangji.ac.kr; 2Department of Computer Engineering, SangJi University, Kwang-Man Ko. 83 Sangjidae-gil, Wonju-si 26339, Republic of Korea; a01014@sj.sangji.ac.kr

**Keywords:** community-dwelling older individuals, comorbidity, deep learning, frailty, survival prediction model

## Abstract

In an aging society, maintaining healthy aging, preventing death, and enabling a continuation of economic activities are crucial. This study sought to develop a model for predicting survival times among community-dwelling older individuals using a deep learning method, and to identify the level of influence of various risk factors on the survival period, so that older individuals can manage their own health. This study used the Korean National Health Insurance Service claims data. We observed community-dwelling older people, aged 66 years, for 11 years and developed a survival time prediction model. Of the 189,697 individuals enrolled at baseline, 180,235 (95.0%) survived from 2009 to 2019, while 9462 (5.0%) died. Using deep-learning-based models (C statistics = 0.7011), we identified various factors impacting survival: Charlson’s comorbidity index; the frailty index; long-term care benefit grade; disability grade; income level; a combination of diabetes mellitus, hypertension, and dyslipidemia; sex; smoking status; and alcohol consumption habits. In particular, Charlson’s comorbidity index (SHAP value: 0.0445) and frailty index (SHAP value: 0.0443) were strong predictors of survival time. Prediction models may help researchers to identify potentially modifiable risk factors that may affect survival.

## 1. Introduction

Population aging is a major problem worldwide, including in Korea. According to the future population projections of Statistics Korea, this country will become a super-aged society by 2025 as the elderly population comprises 20.6% of the total population [1]; Korea is the country with the fastest aging rate in the world [2]. Population aging also affects economic growth due to a decrease in the productive workforce. One way to alleviate the decreasing productive workforce in order to maintain national competitiveness is to include youth under the age of 15 years and/or seniors over the age of 65 years in the workforce. Rather than sending young people under the age of 15 years, who are still immature physically, mentally, emotionally, and socially, into the labor market, retaining individuals older than 65 years who are physically healthy in the productive workforce, and protecting them against early death.

With increasing age, it becomes more important to stay as physically healthy as possible. One study showed that the cost of productivity loss due to premature death of the individuals aged 70 years or older was estimated at KRW 4.5 trillion–KRW 5.4 trillion for men and KRW 1.4–KRW 1.7 trillion for women [3]. All individuals inevitably age, both physically and mentally [4]. Muscle mass decreases, the immune system weakens, and the function of all organs deteriorates [5,6]. Aging is not a disease per se, but as homeostasis and recovery ability decrease [7], the risk of contracting diseases increases, and eventually leads to death. Korea provides free health checkups for transitional stages at the ages of 44 years and 66 years. The National Health Screening Program for Transitional Ages is a nationwide screening program that was started in 2007 and is supported free of charge to the individual by the government, to detect and manage chronic diseases and health risk factors at an early stage at ages when major health changes are likely to occur due to aging and lifestyle [8]. If people know how much a change in a factor could affect a health outcome, due to this life transition period health checkup, they may be able to discard unhealthy behaviors and adopt healthy behaviors themselves, allowing them to manage their own health.

Most previous studies on this topic have shown which factors have an effect on mortality or survival time through survival analysis [9,10,11,12] or logistic regression analysis [12,13,14]. These studies described their results in terms of “people with risk factors have [several times] the risk of death compared to those without them”. The description of the risk of death having doubled or tripled is abstract and does not have clear meaning to most people. We think that the most effective way to encourage self-management is to predict how long a person’s survival period will be, and to indicate how much the survival period will increase if certain factors are changed.

Therefore, the purpose of this study was to construct a model for predicting survival times among community-dwelling older individuals by using a deep learning method, and to identify the level of influence on the survival period according to risk factors, so that older individuals can manage their own health.

## 2. Methods

### 2.1. Data and Study Design

We conducted a deep-learning neural network analysis to build a survival prediction model. To this end, the study made use of the Korean National Health Insurance Service (KNHIS) claims database. This database was constructed from a cohort of 510,000 people (8%) out of approximately 6.4 million individuals, aged 60 years or older in January 2008, who were eligible for national health insurance and medical aid from 2002 to 2019. These individuals were randomly stratified by sex, age, 10th percentile of insurance premium, and regional district [15]. The data have undergone anonymization and de-identification for research use.

The subjects of this study in particular were older people who lived in a community and took part in the National Health Screening Program for Transitional Ages at the age of 66 years, and were followed up for up to 11 years, from 1 January 2009 to 31 December 2019. The National Health Screening Program for Transitional Ages was conducted from 2007, but as checkup items such as activities of daily living (ADL), fall, and urinary discomfort, which are required for calculating the frailty score, were only included from 2009; we targeted people who underwent the national health checkup at the age of 66 years from 2009 to 2019.

The study was conducted in accordance with the Declaration of Helsinki and was approved by the Institutional Review Board of University of Sangji (1040782-221214-HR-15-108).

### 2.2. Study Population

From 2009 to 2019, 258,000 older people received national health screening checkups at the age of 66 years. We calculated the frailty index as described in a previous study [10]. In this study, older individuals who had less than 80% of the information required to calculate the frailty index score were defined as having insufficient data and were excluded from the study. We also used the subject exclusion criteria used in a preceding report: 68,303 were excluded because they had insufficient data to calculate the frailty score. Of the remaining 189,697, 180,235 (95.0%) survived from 2009 to 2019, and 9462 (5.0%) died. Since it is not possible to censor data for those who survived during the follow-up period; as in survival analysis, we used the data of the 9462 deceased as the final dataset for this study (Figure 1).

### 2.3. Variables

The dependent variable of this study was the survival duration in those who died during the follow-up period. Death was defined as all-cause mortality. We used the standard procedure for calculating the frailty index. We could not use the same items used in the operational definition of frailty in Fried et al.’s study [10]. Because we basically used data based on claims data, we could not obtain information such as grip strength. However, efforts were made to include items similar to those that defined frailty in the study, and the definition was made following previous study [16] that used operational definitions based on claims data. Previous studies used 39 health deficit items in five health domains that were assessed during screening examination: medical history (15 items), biometric or laboratory measures (8 items), physical health (2 items), psychological health (8 items), disability (6 items), and chronic conditions (8 items: arthritis, asthma, chronic kidney disease, congestive heart failure, coronary artery disease, chronic obstructive pulmonary disease, type 2 diabetes, and cancer, excluding nonmelanoma skin cancer) [10]. We modified this approach: in previous studies, only eight chronic diseases were included in the frailty index calculation while the scores for chronic conditions were calculated separately as the Charlson’s comorbidity index (CCI) [17]. Because there were many chronic diseases that affect death or survival time, including the eight chronic diseases used to calculate the frailty index, we calculated them as separate risk factors. However, diabetes was included in the CCI calculation, but hypertension and dyslipidemia were not calculated. Diabetes, hypertension, and dyslipidemia are common diseases among the elderly population, and we wanted to reflect hypertension and dyslipidemia, which are not included in the CCI calculation, and also determine how much of an impact the combination of these three diseases has on survival time. In our previous study, the eight chronic conditions were scored as 1 point if the chronic disease was present and 0 points if not present. However, since cancer and diabetes may have different levels of influence on death or health outcomes. The CCI was calculated by weighting each chronic disease. For example, diabetes is counted as 1 point, and diabetes with complication is counted as 2 points. Thus, whereas 39 items were used to calculate the frailty index in a previous study, as the proportion of existing health deficits (range 0–1.00, with higher scores indicating greater frailty), the 8 chronic condition items were excluded from the 39 items in our study, so that 31 items were used to calculate the proportion of existing health deficits (range 0–1.00; Appendix A). Covariates included sex, health insurance status (national health insurance, medical aid), income level (5 categories: quintiles, for which we used insurance premium as a proxy, level of disability (none, mild, severe), grade of long-term care benefit, and admission to an intensive care unit (ICU). The level of disability was determined by the National Pension Service committee, which consisted of two medical specialists and a social worker, and was based on clinical documentation (disability certificate, medical records, and test results) and video evaluation. The level of disability is divided into six grades, and if there is no disability grade, it is classified as “none”. Among those with a disability grade, grades 1 and 2 are classified as “severe”, and the remaining grades are classified as “mild”. Korea provides long-term care benefit to older individuals who have difficulty in taking care of themselves for more than 6 months due to old age or geriatric disease. The extent of long-term care benefit is divided into five grades (grades 1–5). The lower the grade, the more long-term care is needed. In principle, grades 1–2 indicate the state of being eligible for admission to a nursing facility, and grades 3–5 indicate the state of receiving long-term care services at home. Admission to an ICU was defined according to the experience of being admitted to an ICU during the year of transitional life checkup.

### 2.4. Statistical Analysis

The Shapley additive explanations (SHAP) algorithm was applied in this study to calculate how much influence each feature had on the predicted result, based on the Shapley value in game theory (Figure 2). The value obtained by the predictive model indicates the importance of the feature. To obtain SHAP, we used tree-SHAP, which improved the calculation speed by using a tree model, rather than the traditional kernel-SHAP. Among various tree algorithms, XGBoost has a boosting feature and has the advantage of faster learning speed than random forest.

In this paper, the SHAP learning model was applied because the training dataset was very small, and a regression model was used rather than a classification model. In addition, all data were used as training and test data, without distinguishing between training data and test data. In the final stage, values such as 0.0, which were found to be meaningless in the SHAP value, were excluded because they did not affect the learning model.

The model learning time was 482.07 s. The SHAP value was calculated using the learned model and the shapviz library [18]. This process consumed 65.31 s. The input variable (X) of the learning model utilized 10 features, excluding LIFE_TIME data from raw data. The output variable (Y) of the learning model had a value of 0–10.9 as LIFE_TIME (survival time), and min–max scaling was performed in the range of [0, 1] to facilitate model learning. Model learning was accomplished using logistic regression in XGBoost. The learning goal was repeated 1000 times in the direction of reducing the root mean square error (RMSE). In this process, if performance did not improve by more than 50 times, then the process was terminated early. All other parameters maintained the default parameters of XGBoost 1.7.5.1. XGBoostmodeling was conducted with Python 3.8 (Python SoftwareFoundation, Delaware, USA), and other analyses were completed with SAS 9.4 (SAS Institute Inc., Cary, NC, USA).

## 3. Results

Table 1 shows the characteristics of our study’s population. Of the 9462 eligible individuals, 6326 (66.9%) were men and 3136 (33.1%) were women. There were 2436 (25.8%) people with a score of 0 on CCI and 7026 (74.2%) people with a score of 1 point or more. The average frailty index was 0.1547.

In the model learning process, the RMSE by iteration round is represented as a graph, as shown in Figure 3. The performance of the learning model was confirmed with RMSE = 0.1339508 and R^2^ = 0.701071.

Figure 4 shows the predicted survival time (Yp) on the Y-axis and the actual survival time on the X-axis when the learning data were applied to the model. Each point is one input datum. The X value is the actual survival time (LIFE_TIME), and the Y value is the predicted LIFE_TIME. The blue line represents a Y = X graph, where the actual survival time was the same as the maximum predicted by the model. The green line represents the median value of YPred with the same Y, and the red line is the trend line drawn based on these median values. In the graph, the first LIFE_TIME period of 0.38 (4.1 years) or less tended to predict a survival time longer than the actual time. Above 0.38, the predicted survival time was shorter than the actual survival time.

Figure 5 shows the result of calculating the arithmetic average of the absolute SHAP values for each feature, after calculating the SHAP value through the additionally learned model.

Plotting the feature value and the SHAP value of each input datum as a point is shown in Figure 6. The CCI, frailty index, and long-term care benefit grade had the greatest influence, while admission to an ICU had the least influence. Since the predicted value is expressed by adding the SHAP values of all features of the corresponding data, the output (survival period) of the model became smaller when the SHAP value was negative, and became larger when the SHAP value was positive. In the case of long-term care benefit grade, the larger the value, the more negative was the SHAP value and the lower was the survival prediction. In contrast, for CCI and the frailty index, the life expectancy decreased as the value of the feature increased, and the life expectancy increased as the value of the feature decreased.

When the CCI was 0, the SHAP value was about 0.05. As the CCI value increased, the SHAP value decreased. Thus, it can be seen in Figure 7 that the predicted survival time became shorter as the CCI value increased.

## 4. Discussion

In this study we investigated risk factors affecting survival time for older Koreans during an 11-year follow up using a deep-learning-method-derived risk calculator. Unlike previous studies, our study not only discovered factors that affect the survival period, but also allows us to arithmetically calculate how many years a person can survive depending on the presence or absence of factors that affect the survival period if observed for 11 years. It is different from previous studies that simply show results as “which factor has a several times higher risk of death or survival period”. It is easy to understand and can raise awareness.

The degree of influence of each risk factor on the survival period was expressed using the SHAP value. The most influential factor during the 11-year survival period was CCI, which represents chronic diseases in older individuals. Long-term care benefit grade was the third most influential factor. Diabetes, hypertension, and dyslipidemia, which are common among chronic diseases, were treated as separate variables rather than including them in the CCI calculation. The SHAP value of the combination of diabetes, hypertension, and dyslipidemia was about half of that of the CCI. As such, diabetes, hypertension, and dyslipidemia are significant factors that affect survival time as much as other chronic diseases.

In our study, it has been confirmed that both frailty and chronic diseases have a significant impact on survival time. Chronic diseases and frailty are closely related [19,20]. Frailty has been described as the loss of ability to adapt to stress because of diminished functional reserves [21]. Some studies propose that the presence of chronic diseases contributes to the onset of frailty [21,22]. In addition, many studies have found that various chronic diseases can cause frailty, which reduces body function, or that the side effects of medications used to treat chronic diseases can cause frailty [10,22,23,24,25,26,27,28]. Frailty and the deterioration of body functions can lead to death or serious disability. In particular, the SHAP value related to the effect of CCI on survival period indicated that the survival period decreased by 0–2 points according to the comorbid conditions. Since one or two chronic diseases did not reduce survival time, whereas three or more complex chronic diseases decreased survival time, it is important to prevent progression to multiple chronic diseases. If it is possible to prevent progression to complex chronic diseases, then it is possible to prevent senescence. Since most chronic diseases are caused by modifiable risk factors such as lifestyle habits, if the elderly population can manage their health through regular lifestyle habits, exercise, smoking cessation, moderation in drinking, and stress management from a young age, then older people will be able to delay aging and live healthily into their old age.

We confirmed that the frailty index also reduced the survival period once it reached a value of 0.3 or more. In many preceding studies, a frailty index of 0.3 defined transition of the frailty level from moderate to severe. We calculated the frailty index using 31 factors, some of which were modifiable factors. For those who did not apply for long-term care benefit grade, the SHAP value of the survival period had a negative value. Our survival time prediction model was based on individuals who died during the follow-up period, and we found that survival time was positive among those with the grade that required the most long-term care services, whereas the survival time decreased in those who did not apply for a long-term care benefit grade. We consider that this was due to death occurring before these individuals applied for long-term care benefit grade.

Our research had significant limitations. Our survival prediction model was built on a population of less than 10,000 using a deep learning method. Second, we did not consider other factors affecting survival that were not included in the claims database. In addition, a predictive model was implemented using characteristics of the individuals at the age of 66 years, but factors that could change after the age of 66 years, such as the frailty index, chronic disease status, smoking status, and drinking habits, could not be considered. In addition, the frailty index was calculated using 31 items, but these 31 items were also characteristics noted at the time of the life transition period examination at the age of 66 years, and only data at one point in time were used, without reflecting factors that change daily or yearly.

Despite these limitations, our research offers some advantages. The present study constructed a model for predicting the survival period, rather than death or survival, and targeted older individuals living in the community rather than patients with specific diseases and extended the observation over 11years.

## 5. Conclusions

The number of chronic diseases, the frailty index, the long-term care benefit grade, and a combination of diabetes, hypertension, and dyslipidemia were found to be the factors that most affected survival time. As the number of chronic diseases and the frailty index increased, the survival period decreased. Prediction models such as ours may help identify potentially modifiable risk factors that may affect survival.

## Figures and Tables

**Figure 1 geriatrics-08-00105-f001:**
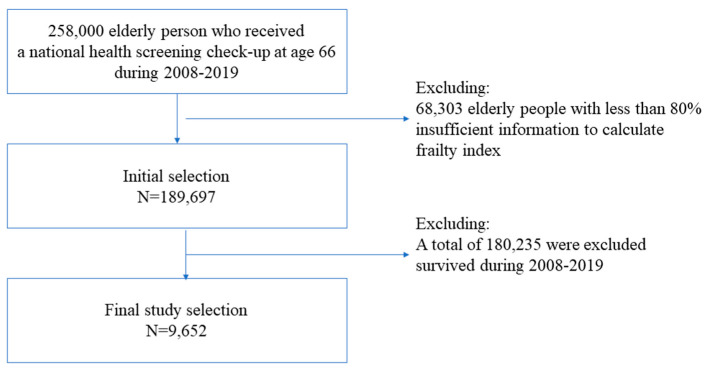
Flowchart for sample selection.

**Figure 2 geriatrics-08-00105-f002:**
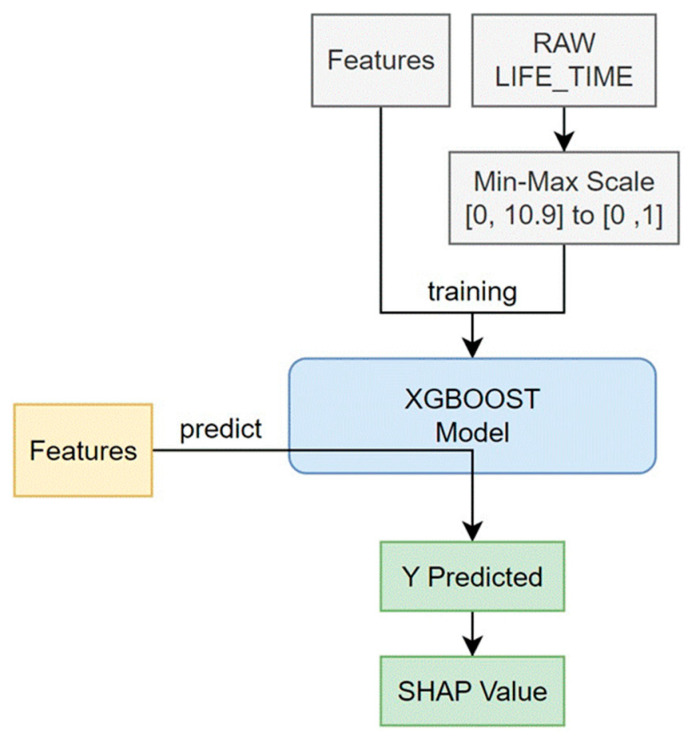
Diagram of the predictive model implementation process. (X) meant 12 features, and (Y) was survival time. The Y-axis is the survival time, and the input data was scaled to min–max in the [0, 1] range to facilitate real data for learning models, and was learned using logistic regression using XGBoost.

**Figure 3 geriatrics-08-00105-f003:**
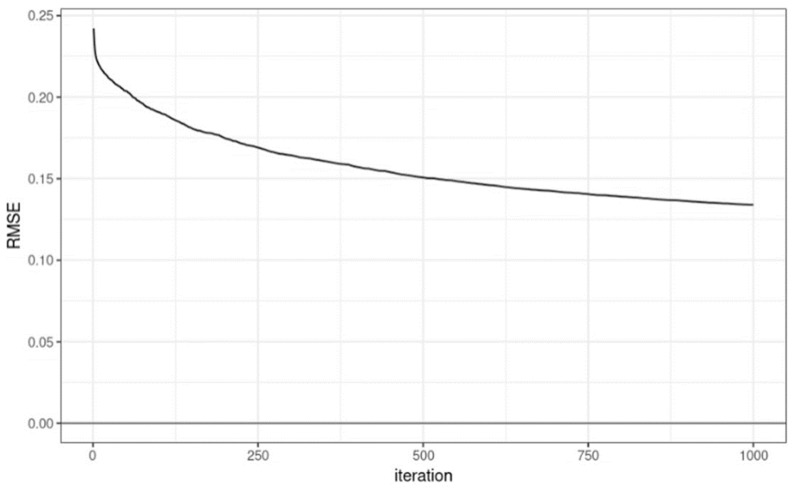
Graph depicting the root mean square error (RMSE) value and iteration number during learning. In this research, we applied the SHAP model, which calculates how much each feature influenced the prediction result and interprets the prediction result based on the importance of the feature.

**Figure 4 geriatrics-08-00105-f004:**
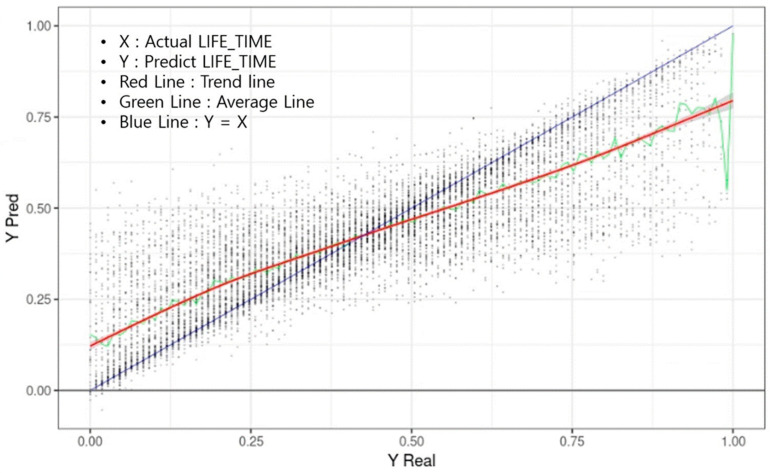
Correlation between actual (blue) and predicted (red) survival time. X-axis means actual survival time; Y-axis means predicted survival time; red line represents trend line; green line represents average line; blue line means Y = X.

**Figure 5 geriatrics-08-00105-f005:**
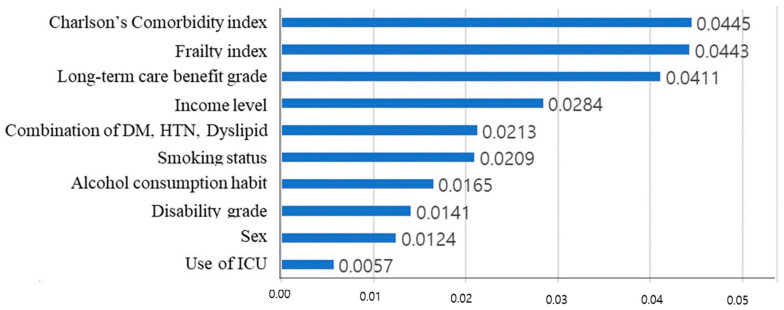
Feature importance in the deep-learning model. Each value represents the mean absolute SHAP value of all data. This figure shows how much each feature affects survival time. The larger the SHAP value, the greater the impact on survival time; CCI, frailty index, long-term care grade influenced most; ICU admission influenced least.

**Figure 6 geriatrics-08-00105-f006:**
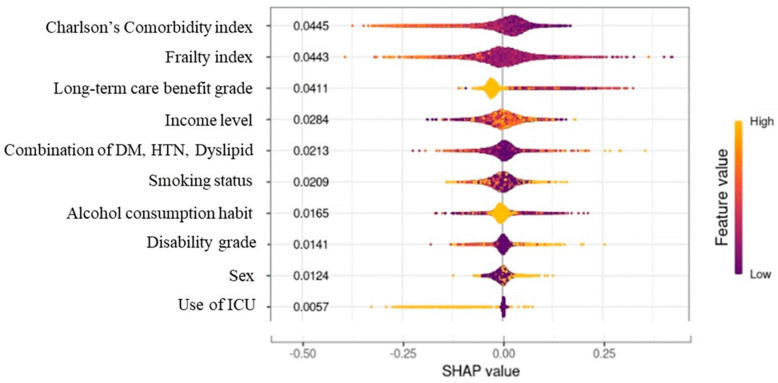
Plotting the feature value and SHAP value. † For each feature, those with high value are displayed in yellow and those with low value are displayed in purple: for example, in the case of sex, men are given a value of 1 and women are given a value of 2, so women are expressed in yellow. The SHAP value, which predicts the survival period for each woman, is plotted. Most yellow dots have positive SHAP values between 0.00 and 025, and most purple dots have SHAP values between 0.00 and −0.125. Having a negative SHAP value means that the survival period is shortened.

**Figure 7 geriatrics-08-00105-f007:**
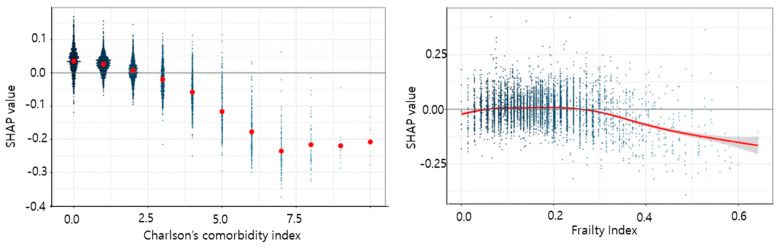
SHAP value according to each feature. The X-axis means the value of the feature, and the Y-axis means the SHAP value according to each feature value. The red dot represents the average SHAP value of each feature value: for example, in the case of CCI, the SHAP values of 2436 people with a CCI value of 0 were plotted, and it can be seen that the average SHAP value of these people is slightly below 0.05 (red dot).

**Table 1 geriatrics-08-00105-t001:** Feature value of study population.

(Feature Value) Individual Variables		
Sex, *n* (%)		
(1): Men	6326	(66.9)
(2): Women	3136	(33.1)
Income level, *n* (%)		
(1): 0 percentile (Medical Aid)	943	(10.0)
(2): 1~20 Percentile	1579	(16.7)
(3:) 21~40 Percentile	1219	(12.9)
(4): 41~60 Percentile	1542	(16.3)
(5): 61~80 Percentile	2115	(22.4)
(6): 81~100 Percentile	2064	(21.8)
Charlson’s comorbidity index, *n* (%)		
0	2436	(25.8)
1	2791	(29.5)
2	1891	(20.0)
3	1119	(11.8)
4	541	(5.7)
5	289	(3.1)
6	220	(2.3)
7	111	(1.2)
8	48	(0.5)
9	11	(0.1)
10 or more	5	(0.1)
Frailty index, Mean (SD)	0.1547	(0.0824)
Long-term care benefit grade, *n* (%)		
1: 1st grade	471	(5.0)
2: 2nd grade	1020	(10.8)
3: 3~5th grade	70	(0.7)
4: Out of grade	463	(4.9)
5: Those who have not applied for a grade	7438	(78.6)
Disability grade, *n* (%)		
(1): 1 None	7384	(77.7)
(2): Severe	872	(9.2)
(3): Mild	1236	(13.1)
Combination of DM, HTN, and dyslipidemia, *n* (%)		
(0): DM(+), HTN(−), dyslipidemia(−)	5660	(59.8)
(1): DM(−), HTN(+), dyslipidemia(−)	2385	(25.2)
(2): DM(−), HTN(−), dyslipidemia(+)	41	(0.4)
(3): DM(+), HTN(+), dyslipidemia(−)	778	(8.2)
(4): DM(+), HTN(−), dyslipidemia(+)	42	(0.4)
(5): DM(−), HTN(+), dyslipidemia(+)	528	(5.6)
(6): DM(+), HTN(+), dyslipidemia(+)	12	(0.1)
(7): DM(−), HTN(−), dyslipidemia(−)	16	(0.2)
Smoking status, *n* (%)		
(0): No smoking	4910	(52.0)
(1): Ex-smoking	2166	(22.9)
(2): Current smoking	2374	(25.1)
Alcohol consumption habit, *n* (%)		
(0): No drinking	870	(9.2)
(1): 2~3 times per month	1299	(13.7)
(2): Once or twice per week	454	(4.8)
(3): More than 3 times per week	6839	(72.3)
Use of Intensive care unit, *n* (%)		
(0): No	9232	(97.6)
(1): Yes	230	(2.4)

## Data Availability

These data are provided only to researchers, and researchers can also access the data using a security key. According to the National Health Insurance Service’s data disclosure regulations, the data cannot be disclosed to anyone other than researchers.

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
