# Peer review of "Long-Term Survival Prediction Model for Elderly Community Members Using a Deep Learning Method"

_geriatrics, 2023, doi:10.3390/geriatrics8050105_

Round 1

Reviewer 1 Report

I read the article with great interest. I would like to see the following points corrected for the better understanding of a wide range of readers.

Major points

The authors conclude that the number of chronic diseases, frailty index, long-term care benefit grade, and the combination of diabetes, hypertension, and dyslipidemia were found to be the factors most influential on survival. Please explain the novelty of this result in comparison to previous studies. Also, are diabetes and hypertension not included in the number of chronic diseases? Please clarify the definition of chronic diseases. Similarly, I cannot deny the impression that the definition of frailty is vague. The phenotype model used by Fried et al. in the Cardiovascular Health Study (CHS) (doi: 10.1093/gerona/56.3.m146.) is used worldwide in the definition and assessment scale of physical frailty. We believe it is necessary to carefully explain why the authors used the rating scale used in the present study instead of this CHS. These are the most important points in this study, so if you have a chronic disease, use the WHO definition (https://www.who.int/publications/i/item/multimorbidity.), and if you are frail, use the above definition or Please carefully mention the differences with the rating scale. After that, I would like to perform statistical analysis.

The World Health Organization's definition of elderly is 65 years of age or older. However, in this study, the age group of 60 years and older is considered "elderly". This rationale needs to be clarified and a method presented.

Minor points

The words "men", "male", "women", and "female" are mixed up. Unification is needed.

Author Response

Thank You very much for your comments. In our revised manuscript, we did our best to incorporate your comments.

Reviewer 2 Report

This manuscript represents description of a risk prediction model for older adults.

Recommendations for enhanced clarity are provided to the authors and the attached document.

Author Response

We changed as your comments. I would appreciate it if you could check it in the paper.

Round 2

Reviewer 1 Report

Thank you for your careful revisions.

I have the impression that it is easier to read than last time.

I would like you to correct the following

Words such as elderly people, elderly person and older people are mixed up.

They need to be unified.

Author Response

Thank you very much for your comments. In our revised manuscript, we changed as your comment. We unified as older people. 

Reviewer 2 Report

The authors have made a credible response to reviewer comments.

Suggest delete line 416 as repetitive.

Suggest rephrase line 416: ...and extended the observation over 11 years.

NA

Author Response

Thank you very much for your comments. In our revised manuscript, we rephrased line 416.